# Training Differentially Private Graph Neural Networks with Random Walk Sampling

**Morgane Ayle**
Technical University of Munich
`morgane.ayle@tum.de`

**Jan Schurchardt**
Technical University of Munich
`schuchaj@in.tum.de`

**Lukas Gosch**
Technical University of Munich
`lukas.gosch@in.tum.de`

**Daniel Zügner**
Technical University of Munich
`zuegnerd@in.tum.de`

**Stephan Günnemann**
Technical University of Munich
`guennemann@in.tum.de`

## Abstract

Deep learning models are known to put the privacy of their training data at risk, which poses challenges for their safe and ethical release to the public. Differentially private stochastic gradient descent is the de facto standard for training neural networks without leaking sensitive information about the training data. However, applying it to models for graph-structured data poses a novel challenge: unlike with i.i.d. data, sensitive information about a node in a graph cannot only leak through its gradients, but also through the gradients of all nodes within a larger neighborhood. In practice, this limits privacy-preserving deep learning on graphs to very shallow graph neural networks. We propose to solve this issue by training graph neural networks on disjoint subgraphs of a given training graph. We develop three random-walk-based methods for generating such disjoint subgraphs and perform a careful analysis of the data-generating distributions to provide strong privacy guarantees. Through extensive experiments, we show that our method greatly outperforms the state-of-the-art baseline on three large graphs, and matches or outperforms it on four smaller ones.

## 1  Introduction

The introduction of Graph Neural Networks (GNNs) has enabled the training of Deep Learning (DL) models on graph-structured data and for various tasks such as node classification, link prediction or graph classification. However, similar to DL models trained on image [9] or text data [11, 2], GNNs leak information about their training data [20, 14, 22], such as the features of a node, or which nodes are connected by an edge.

In this paper, we analyze the privacy of GNNs under the lens of Differential Privacy (DP) [8]. In particular, we ensure the privacy of all nodes' features in a graph. While DP-SGD [1] is the de facto standard for training DL models with DP, its transfer to GNNs is not straightforward given the non-i.i.d. nature of the data. Indeed, since an $L$-layer GNN typically uses the $L$-hop neighborhood of a node during the forward pass, the gradient of a node does not depend on that node alone, but on all nodes in its neighborhood. While some works [7, 16] have attempted to apply DP to GNNs, most of them focus on edge-level DP. Methods that can be applied to feature-level DP suffer from

2022 Trustworthy and Socially Responsible Machine Learning (TSRML 2022) co-located with NeurIPS 2022.

loose privacy guarantees [7], or rely on custom GNN architectures [16]. We propose an adaptation of DP-SGD to train GNNs with feature-level DP while attenuating the aforementioned problem and preserving a high model utility. We experimentally demonstrate that our method can offer significantly stronger privacy guarantees than prior work, particularly on large graphs.

## 2 Background

### 2.1 Differential privacy

**$(\epsilon, \delta)$-DP** Differential Privacy (DP) [8] is a notion of privacy that allows data analysts to extract useful statistics from a dataset, without leaking too much information about the samples in it. More formally, given two neighboring datasets $D$ and $D'$ – denoted $D \sim D'$ – that differ by one sample (either by deleting, adding or modifying a sample), a randomized algorithm $\mathcal{M}$ with co-domain $Y$ is $(\epsilon, \delta)$-DP if for all $O \subseteq Y$, and for all $D \sim D'$, $Pr[\mathcal{M}(D) \in O] \leq \exp(\epsilon)Pr[\mathcal{M}(D') \in O] + \delta$. The parameters $\epsilon$ and $\delta$ are the privacy budget parameters: the smaller their values, the better the privacy guarantees.

**$(\alpha, \gamma)$-RDP** An alternative definition of DP is Rényi Differential Privacy (RDP) [12]. A randomized algorithm $\mathcal{M}$ is said to be $\gamma$-RDP of order $\alpha$ – or $(\alpha, \gamma)$-RDP – if for any $D \sim D'$ it holds that $D_\alpha(\mathcal{M}(D), \mathcal{M}(D')) \leq \gamma$, where $D_\alpha = \frac{1}{\alpha-1} \log \mathbb{E}_{x \sim Q} \left( \frac{P(x)}{Q(x)} \right)^\alpha$ is the Rényi divergence of order $\alpha$ which measures the similarity of the distributions $P$ and $Q$. Note that if $\mathcal{M}$ is $(\alpha, \gamma)$-RDP, then it is also $(\epsilon, \delta)$-DP for any $0 < \delta < 1$ where $\epsilon = f_{\text{RDP} \to \text{DP}}(\alpha, \gamma, \delta) = \gamma + \log(\frac{\alpha-1}{\alpha}) - \frac{\log \delta + \log \alpha}{\alpha-1}$ [4]. We rely on $(\alpha, \gamma)$-RDP during our analysis, but report our results in terms of $(\epsilon, \delta)$-DP following prior work.

**The Gaussian mechanism** Given an algorithm $\mathcal{A}$ with real-valued output space $\mathcal{A} : \mathbb{N}^\mathcal{D} \to \mathbb{R}^d$, the Gaussian mechanism privatizes the algorithm by adding Gaussian noise to the outputs of $\mathcal{A}$, i.e. $\mathcal{M} = \mathcal{G}_\sigma(\mathcal{A}(D)) = \mathcal{A}(D) + \mathcal{N}(0, \sigma^2)$. Given that the $\ell_2$ sensitivity of $\mathcal{A}$ is $\Delta_2 \mathcal{A}(D) = \max_{D \sim D'} \|\mathcal{A}(D) - \mathcal{A}(D')\|_2$, the mechanism satisfies $(\alpha, \gamma(\alpha))$-RDP, with $\gamma(\alpha) = \frac{\alpha(\Delta_2 \mathcal{A})^2}{2\sigma^2}$. Intuitively, this indicates that the larger the sensitivity of the function, the more noise needs to be added to obtain a small privacy budget, and therefore the worse the final performance will be. A small sensitivity is therefore desirable.

**Amplification by sub-sampling** A useful property of DP (and RDP) is that, given a mechanism $S$ that samples a sub-set of the dataset $D$, applying a private mechanism to $S(D)$ leads to better privacy guarantees than applying it to the entire dataset $D$. Intuitively, this is due to the fact that subsampling introduces a non-zero chance of an added or modified sample to not be processed by the randomized algorithm. Typically, $S$ is assumed to be a Poisson or uniform sampling over the dataset. Poisson sampling is typically used when the neighboring datasets differ in size, while uniform sampling is used otherwise. In this paper, we rely on uniform sampling.

### 2.2 Differential privacy in deep learning

Differentially Private Stochastic Gradient Descent (DP-SGD) [18, 5, 1] is the foundation of many works [7, 11, 10] that apply DP to deep learning. It privatizes the weights of a model with respect to the input dataset at every iteration of training, and then accumulates the privacy budget being spent over all iterations. One private training iteration consists of batching a set of samples, computing the gradient on each sample independently, clipping the norm of each gradient vector to a maximum norm $C$, calculating the entire gradient by adding calibrated Gaussian noise, and finally performing an update step. The clipping step is used to bound the sensitivity of the gradients to changes in the input. Then, assuming that two neighboring datasets $D$ and $D'$ differ in the features of one sample, the sensitivity of the total gradient on a batch of i.i.d. samples is bounded by $2C$. Through batching (i.e. sub-sampling the dataset using a sampling mechanism $S$), amplification by sub-sampling theorems [3, 19] can be exploited to get better privacy guarantees at every iteration. Finally, assuming each iteration $t$ is $(\alpha, \gamma_t)$-RDP, the overall training is then $(\alpha, \sum_{t=0}^{T} \gamma_t)$-RDP [12] where $T$ is the total number of iterations.

## 2.3 Graph neural networks

**Definition**    In the following, we define a graph as $G = \{X, A\}$, where $X \in \mathbb{R}^{N \times d}$ is the feature matrix in which each row corresponds to one node's feature vector, and $A \in \{0, 1\}^{N \times N}$ is the adjacency matrix in which $A_{ij}$ is 1 if there exists an edge between nodes $i$ and $j$ and 0 otherwise. Note that we only consider undirected graphs, therefore $A = A^T$. Graph Neural Networks (GNNs) are a class of models that learn a mapping $f : G \to Z \in \mathbb{R}^{N \times d'}$, where $Z$ is an updated feature matrix of $G$ that can be used for various downstream tasks. Each layer of a GNN typically consists of two steps: 1) in the aggregation step, information about the neighborhood of every node is gathered; 2) in the update step, the feature vector of every node is updated based on its current feature vector and the aggregated neighborhood information.

**The receptive field**    The receptive field of a node in a GNN is defined as the region in the input graph that influences the GNN's predictions for that specific node. For a GNN with $L$ layers, the receptive field of a node $v$ is the $L$-hop neighborhood of $v$. Thus, for a graph with maximum node degree $K$, the largest possible receptive field size of any node $v$ is $\mathrm{RF}(v) = \sum_{l=0}^{L} K^l = \frac{K^{L+1}-1}{K-1}$, i.e. the receptive field grows exponentially with the number of layers of the GNN.

## 2.4 Differential privacy in graph neural networks

Given that graphs contain two types of attributes – node features and edges – multiple levels of DP [6, 7, 16] can be considered: *edge-level* DP, where the edges between nodes are private; *feature-level* DP, where the features of nodes are private; and *node-level* DP, where both the features and edges of nodes are private. In this work, we focus on feature-level DP using DP-SGD. Contrary to traditional i.i.d. datasets, samples in a graph (i.e. nodes) are not independent: changing the features of one node affects the gradients of all nodes within the receptive field of the modified node. In fact, the sensitivity of the total gradient on a graph is bounded by $2\frac{K^{L+1}-1}{K-1}C$ (see Appendix A), which grows exponentially with the number of layers $L$. Given that the Gaussian mechanism adds noise proportional to the sensitivity of the total gradient, this can lead to large amounts of noise being added during training, which in turn leads to poor final model utility.

## 3    Related work

In [15], a node-level differentially private GNN is trained by perturbing features and edges locally before sending them to a global server. This setup is called local DP, and differs from our notion of DP where a central learner is trusted with the real data. The authors in [10] propose to split the graph into disjoint sub-graphs using uniform node sampling, then treat each sub-graph as an independent sample. Note that, contrary to our method which considers privacy at the individual node feature level, their approach treats the entire graph as a datapoint to privatize, rather than providing privacy for the individual nodes in the graph. The method in [16] privatizes GNNs at both the node-level and edge-level. However, their approach only applies to the GNN architecture they propose and not to arbitrary GNNs, unlike our proposed method. Furthermore, it does not resolve the issue of exponentially growing sensitivity in transductive learning scenarios. For a survey on DP on graph data, refer to [13]. Finally, the authors of [7] propose to reduce the sensitivity of a GNN's gradients by bounding the maximum degree $K$ of the graph. However, this does not resolve the exponential growth with the number of layers. Therefore, they still obtain loose privacy guarantees ($\epsilon = 20$). Since this method is the closest to our setup, we compare our approach to theirs in our experiments.

## 4    Methodology

### 4.1    Approach

We propose to adapt DP-SGD to the graph domain to ensure that the weights of a GNN are private with respect to the nodes' features, while overcoming the problem of requiring exponentially more noise with a growing network depth. In the following, we define two graphs $G$ and $G'$ as neighbors if they share the same structure $A$ and number of nodes $N$ but differ in one row of the feature matrix $X$ corresponding to the modified node $\tilde{v}$. We want to train the GNN such that for all $G \sim G'$,

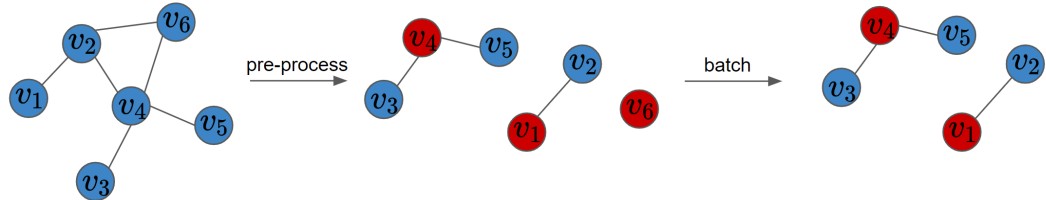

Figure 1: Our general sampling method. Starting with a graph, we generate subgraphs by first sampling a root node (depicted in red), and then sampling one or more random walks starting from the root node. Every node appears in exactly one subgraph. Before every iteration, we batch $m$ many subgraphs, where $m = 2$ in this case. Root nodes are used as training nodes, while remaining nodes are used for aggregation in the GNN only.

$D_\alpha(\mathcal{M}(G), \mathcal{M}(G')) \leq \gamma$, where $\mathcal{M}$ is a randomized algorithm that returns the weights of the GNN.

To adapt DP-SGD to the graph domain, we propose to pre-process the graph into sets of independent subgraphs that do not affect each others' gradients, so that the sensitivity of the total gradient on any batch depends on the gradient of one subgraph only. We summarize our training procedure in Algorithm 1. More precisely, we pre-process the graph into a set of $M$ disjoint subgraphs $G_S = \{s_1, s_2, \ldots, s_M\}$, i.e. subgraphs that do not have any nodes in common, using sampling method $S$. Each subgraph $s_i$ consists of two components: 1) one training node $v_i$, and 2) a set of neighbors $\mathcal{N}(v_i)$ that is used for the aggregation step of the GNN. At training time, for every iteration $t$, we create a batch by sampling $m$ subgraphs uniformly at random from the set of subgraphs $G_S$. We then compute the gradients $\nabla_{\mathbf{w_t}} \mathcal{L}(v_j, \mathcal{N}(v_j))$ on all training nodes and clip the norm of each to a value C. We compute the total gradient by summing individual gradients and adding Gaussian noise. Finally, we update the weights.

Due to the disjointness of subgraphs, changing one node's features – whether it is a training node or a neighbor – will affect at most one subgraph (i.e. sample) in the batch, which reduces the upper bound on the sensitivity of the total gradient to $2C$. Since we sample subgraphs uniformly at random, we can leverage the strong amplification by sub-sampling theorem [19], i.e. account for the possibility of the gradient not being affected if the modified node $\tilde{v}$ is not part of the batch.

We generate these disjoint subgraphs via random walk sampling, which is an effective way of training GNNs [21]. We choose random walk sampling, since it ensures that nodes form a connected subgraph of a training node's neighborhood, while limiting the number of nodes being sampled from that neighborhood (i.e. from the receptive field). In the following, we propose three different random-walk-based sampling methods, which we later compare in our experimental results. Furthermore, we derive for each sampling method a tight upper bound on the probability of sampling the modified node $\tilde{v}$ in a batch, which is required for applying the amplification by subsampling theorem in [19].

## 4.2 Sampling methods

Our three sampling methods consist of pre-processing the graph into a set of $M$ disjoint subgraphs $G_S = \{s_1, s_2, \ldots, s_M\}$, and then generating a batch $B \subseteq G_S$ by sampling $m$ subgraphs uniformly at random. An overview of our general approach is depicted in Figure 1. Given a graph with $M$ generated disjoint subgraphs, the true probability of sampling node $\tilde{v}$ is $P[\tilde{v}] = \frac{1}{M}$, since we know that a node is in exactly one of the $M$ subgraphs. However, to ensure differential privacy, we require a bound that holds for all possible graphs and any run of the sampling procedure. Thus, we use the upper bound $P[\tilde{v}] = \frac{1}{M} \leq \frac{1}{M_{\min}}$ where $M_{\min}$ is the minimum number of subgraphs that can be generated in any graph of $N$ nodes. Then, the probability of sampling $\tilde{v}$ in a batch of $m$ subgraphs using sampling mechanism $S$ is at most $P_S[\tilde{v}] \leq \frac{m}{M_{\min}}$.

**Algorithm 1** DP-SGD with random walk sampling

---

**Input:** Graph $G = \{V, E\}$, sampling method $S$, loss function $\mathcal{L}$, initial model weights $\mathbf{w_0}$, noise standard deviation $\sigma$, gradient clipping norm $C$, number of iterations $T$, frequency at which to re-sample subgraphs in DRW-D $i$

$G_S = S(G)$            ▷ Generate subgraphs from graph G using sampling method S

**for** t in [0, T) **do**

    **if** t % i == 0 and S == DRW-D **then**

        $G_S = S(G)$

    **end if**

    Sample $m$ subgraphs uniformly at random from $G_S$ to form batch $B$

    **for** $s_j$ in $B$ **do**                      ▷ $s_j$ is a subgraph

        Compute $\nabla_{\mathbf{w_t}} \mathcal{L}(v_j, \mathcal{N}(v_j))$

        $g_t(v_j) = \text{clip}\left(\nabla_{\mathbf{w_t}} \mathcal{L}\left(v_j, \mathcal{N}(v_j)\right), C\right)$      ▷ Compute and clip individual gradients in B

    **end for**

    $g_t(B) = \frac{1}{|B|}\left(\left(\sum_{s_j \in B} g_t(v_j)\right) + \mathcal{N}(0, \sigma^2)\right)$      ▷ Add noise to the gradients

    $\mathbf{w}_{t+1} = \text{update}(\mathbf{w}_t, g_t(B))$      ▷ Update weights based on optimizer being used

**end for**

---

**Disjoint random walks**    The first sampling method we propose is called Disjoint Random Walks (DRW). We pre-process the graph once before training and then generate batches at every iteration using the same set of subgraphs. Each subgraph consists of one random walk of length $L$ (refer to Appendix B for a pseudo-code). A random walk of length $L$ contains at most $L + 1$ nodes, and generating random walks that all have maximal length would result in the minimum number of random walks, since a node can only appear in one random walk. Therefore, we get $M_{\min} = \lceil \frac{N}{L+1} \rceil$ and $P[\tilde{v}] \leq \frac{1}{\lceil \frac{N}{L+1} \rceil}$. Finally, the upper bound probability of sampling a node $\tilde{v}$ is $P_{\text{DRW}}[\tilde{v}] \leq \frac{m}{\lceil \frac{N}{L+1} \rceil}$.

**Disjoint random walks with restarts**    To create better subgraphs that contain more nodes for aggregation, we also propose Disjoint Random Walks with Restarts (DRW-R). Similary to DRW, this sampling method generates subgraphs once before training by using random walks, but instead of sampling one random walk per training node we sample $R$ of them (refer to Appendix B for a pseudo-code). Given a random walk length of $L$ and $R$ restarts, the minimum number of subgraphs is $M_{\min} = \lceil \frac{N}{1 + R \times L} \rceil$ where $1 + R \times L$ is the maximum size of one subgraph when all random walks have length $L$, and the probability of sampling node $u$ in a batch of size $m$ is therefore $P_{\text{DRW-R}}[u] \leq \frac{m}{\lceil \frac{N}{1 + R \times L} \rceil}$.

**Disjoint random walks with dynamic re-sampling**    Finally, we propose a third sampling method in which we pre-process the graph into disjoint subgraphs every $i^{th}$ iteration instead of once before training, where $i$ is a hyper-parameter that is chosen based on the cost of the sampling procedure on each dataset. This allows us to increase the diversity of subgraphs used for training, and prevent overfitting on the subgraphs generated in one run of the sampling procedure. We call this procedure DRW-D, where D stands for Dynamically re-sampling random walks. The probability of sampling node $\tilde{v}$ is the same as in DRW, namely $P_{\text{DRW-D}}[\tilde{v}] = P_{\text{DRW}}[\tilde{v}] \leq \frac{m}{\lceil \frac{N}{L+1} \rceil}$. Note that this method consists simply of re-running the subgraph generation process DRW at every $i^{th}$ iteration instead of once before training, which is reflected in Algorithm 1.

# 5 Experimental results

**Experimental setup**    We report our results on seven datasets, both in the transductive and the inductive settings. The dataset sizes in terms of total nodes range from small (Cora [17], Citeseer [17]) to medium (PPI [21], Pubmed [17]) to large (Flickr [21], Arxiv [21], Reddit [21]), or in number of training nodes from small (Pubmed, Citeseer, Cora) to medium (PPI) to large (Flickr, Arxiv, Reddit). We report the exact number of nodes as well as some additional dataset characteristics in Appendix C. We focus on the node classification task, and report our results in terms of F1 Micro score, a metric equivalent to accuracy except on PPI which is a multi-label classification task. Following prior work, we report our privacy budget using $\epsilon$ and a fixed $\delta$ per dataset (see Appendix

C). Given a target $\epsilon$, we keep training while tracking the $(\alpha, \gamma_t)$ privacy budget being spent until we reach $\epsilon = f_{\text{RDP}\to\text{DP}}(\alpha, \sum_{t=0}^{T'} \gamma_t, \delta)$ at iteration $T'$.

We compare our proposed methodology with each sampling method to three baselines: 1) A basic GCN trained with random walk sampling; 2) A basic MLP trained with uniform node sampling; and 3) The method proposed in [7] which we call FDP for Feature-level DP. Note that while they train their models up to an $\epsilon$ of 20, we only train them until $\epsilon = 8$, since a very large $\epsilon$ does not have much value in terms of privacy.

Table 1: Comparison between the F1 Micro score (%) achieved by a basic GCN and MLP, the FDP baseline, and our proposed method with multiple sampling methods. All DP methods are trained with a target budget of $\epsilon \leq 8$.

| | | Layers | Width | Dataset | | | | | | |
| | | | | Cora | CiteSeer | PPI | PubMed | Flickr | Arxiv | Reddit |
|---|---|---|---|---|---|---|---|---|---|---|
| | GCN (non-DP) | 1 | - | 69.8 | 59.5 | 46.2 | 68.7 | 45.6 | 59.7 | 92.5 |
| | | 2 | 256 | 77.3 | 63.7 | 58.9 | 72.9 | 51.3 | 69.1 | 94.7 |
| | | | 512 | 76.6 | 62.2 | 60.7 | 72.9 | 51.3 | 69.5 | 94.7 |
| | MLP (non-DP) | 1 | - | 43.0 | 37.6 | 45.2 | 61.3 | 45.7 | 52.3 | 67.7 |
| | | 2 | 256 | 47.3 | 36.1 | 52.1 | 61.5 | 36.2 | 52.6 | 69.8 |
| | | | 512 | 44.8 | 39.3 | 53.6 | 63.3 | 38.4 | 52.0 | 69.7 |
| | FDP (DP) | 1 | - | 17.1 | 17.5 | 38.4 | 39.6 | 33.6 | 43.8 | 56.7 |
| | | 2 | 256 | 17.6 | 21.5 | **40.7** | 41.4 | 42.5 | 31.9 | 43.7 |
| | | | 512 | 23.2 | **22.1** | 40.0 | 41.2 | 42.4 | 30.2 | 42.3 |
| Ours | DRW (DP) | 1 | - | 19.9 | 20.6 | 40.2 | **41.7** | 42.1 | 59.2 | 81.4 |
| | | 2 | 256 | 17.2 | 20.9 | 38.7 | 40.3 | **48.7** | 59.6 | 80.2 |
| | | | 512 | 24.9 | 21.3 | 37.9 | 41.1 | 47.9 | 59.2 | 81.8 |
| | DRW-D (DP) | 1 | - | 19.8 | 20.6 | 40.1 | **41.7** | 42.2 | 59.2 | 81.4 |
| | | 2 | 256 | 17.2 | 21.3 | 38.6 | 40.2 | 48.5 | **59.7** | 80.2 |
| | | | 512 | **25.0** | 21.7 | 37.9 | 41.2 | 47.8 | 59.3 | 81.5 |
| | DRW-R (DP) | 1 | - | 18.3 | 19.2 | 40.0 | 40.3 | 42.3 | 59.1 | 82.0 |
| | | 2 | 256 | 17.3 | 20.7 | 38.2 | 40.4 | 48.3 | **59.7** | 81.0 |
| | | | 512 | 24.5 | 21.3 | 36.9 | 40.4 | 48.5 | 59.4 | **82.2** |

**Discussion** Table 5 summarizes our results. A GCN trained without DP always outperforms the ones trained with DP, which is expected since clipping gradients and especially adding Gaussian noise decreases the utility of the final model. However, in some cases our method can almost match the utility of the basic GCN, whereas the FDP baseline struggles. For example, DRW sampling on Flickr can reach up to 48.7% accuracy – which corresponds to 95% of the baseline GCN's performance – whereas FDP reaches only 42.5% accuracy – which corresponds to 83% of the baseline GCN's performance. Similarly, our method achieves 87% of the GCN's performance on the challenging dataset Reddit, while FDP can only reach 60% of the GCN's performance. This shows that our sub-sampling approach is effective at solving the exponential growth of the receptive field while approaching the utility of the non-DP GCN baseline, which makes our method attractive for real world applications. That being said, our method uses a smaller amount of training nodes than what is available at every iteration, even when computational complexity is not an issue (i.e. on small graphs). The effect of this reduction in training training samples is exacerbated on small graphs that do not require batching in non-DP training, which leads to our method performing on-par with the FDP baseline on small datasets.

**Comparison with variable privacy budget** Finally, in Figure 2 we expand on our previous results by reporting the accuracy at various $\epsilon$ checkpoints during training. We report the best results that our method achieved across all sampling methods and compare to the FDP baseline. On all datasets, our method largely outperforms FDP across multiple epsilon values. Moreover, FDP cannot achieve an epsilon lower than 2, whereas our method does while sometimes outperforming FDP at higher privacy budgets.

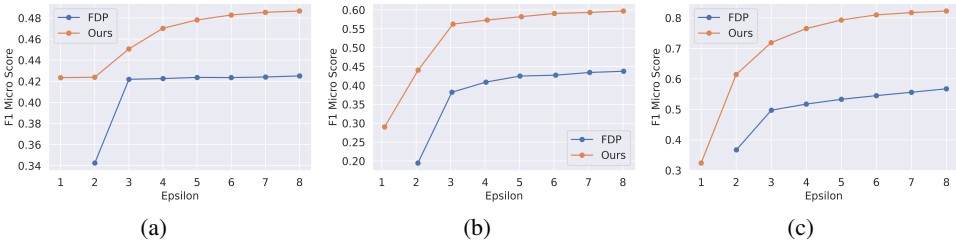

Figure 2: F1 Micro Score vs. epsilon achieved by FDP and our best sampling method for a) Flickr, b) Arxiv and c) Reddit datasets.

# 6 Conclusion

We proposed a novel way of training differentially private graph neural networks. Since graphs consist of inter-connected nodes that influence each other's gradients during training, naively adapting traditional DP methods to graph neural networks can result in unnecessarily large amounts of noise being added to the model during training, which in turn leads to poor utility of the model. We proposed an adapted version of DP-SGD that uses random-walk based sub-sampling to overcome this problem and introduced three sampling methods that generate disjoint subgraphs. For each sampling method, we derived an upper bound on the probability of sampling a modified node in a batch to apply the amplification by sub-sampling theorem and obtain tighter privacy guarantees. Our method achieves a better privacy-utility trade-off compared to the state-of-the-art baseline FDP across multiple datasets, especially for large datasets. A necessary future work direction in this field is to attempt to solve the performance issue on small datasets, which is especially exacerbated on GNNs. For example, pre-training the models on public datasets [11] or using variable signal-to-noise ratios during training are ways of improving the utility in DP. Moreover, different sampling methods that do not necessarily focus on random walks can be explored.

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

## A  Upper Bound on Gradient Sensitivity

We show how to derive the upper bound on the sensitivity of the total gradient on a batch, where $g_t$ is the function that takes a batch $B$ as input and returns the gradients at iteration $t$, $B$ and $B'$ are neighboring batches that differ by one sample $\tilde{v}$, $\mathcal{L}_v$ is the loss function on a node $v$, $\nabla_w \mathcal{L}_v$ is the gradient of the loss on $v$ with respect to the weights of the model, and $\mathcal{I}[\tilde{v} \in B]$ is the indicator function which is 1 if $\tilde{v}$ is in the batch and 0 otherwise.

$$
\begin{aligned}
\Delta_2 g_t &= \|g_t(B) - g_t(B')\|_2 \\
&= \|\sum_{v \in B} \nabla_{\mathbf{w}} \mathcal{L}_v - \sum_{v \in B'} \nabla_{\mathbf{w}} \mathcal{L}_v\|_2 \\
&= \|(\nabla_{\mathbf{w}} \mathcal{L}_{\tilde{v}} + \sum_{u \in RF(\tilde{v}) \setminus \tilde{v}} \nabla_{\mathbf{w}} \mathcal{L}_u) \mathcal{I}[\tilde{v} \in B] - (\nabla_{\mathbf{w}} \mathcal{L}_{\tilde{v}'} + \sum_{u \in RF(\tilde{v}') \setminus \tilde{v}'} \nabla_{\mathbf{w}} \mathcal{L}_u) \mathcal{I}[\tilde{v}' \in B']\|_2 \\
&\leq \|(\nabla_{\mathbf{w}} \mathcal{L}_{\tilde{v}} + \sum_{u \in RF(\tilde{v}) \setminus \tilde{v}} \nabla_{\mathbf{w}} \mathcal{L}_u) \mathcal{I}[\tilde{v} \in B]\|_2 + \|(\nabla_{\mathbf{w}} \mathcal{L}_{\tilde{v}'} + \sum_{u \in RF(\tilde{v}') \setminus \tilde{v}'} \nabla_{\mathbf{w}} \mathcal{L}_{\tilde{v}'}) \mathcal{I}[\tilde{v}' \in B']\|_2 \\
&\leq \|\nabla_{\mathbf{w}} \mathcal{L}_{\tilde{v}} + \sum_{u \in RF(\tilde{v}) \setminus \tilde{v}} \nabla_{\mathbf{w}} \mathcal{L}_u\|_2 + \|\nabla_{\mathbf{w}} \mathcal{L}_{\tilde{v}'} + \sum_{u \in RF(\tilde{v}') \setminus \tilde{v}'} \nabla_{\mathbf{w}} \mathcal{L}_u\|_2 \\
&\leq \|\nabla_{\mathbf{w}} \mathcal{L}_{\tilde{v}}\|_2 + \sum_{u \in RF(\tilde{v}) \setminus \tilde{v}} \|\nabla_{\mathbf{w}} \mathcal{L}_u\|_2 + \|\nabla_{\mathbf{w}} \mathcal{L}_{\tilde{v}'}\|_2 + \sum_{u \in RF(\tilde{v}') \setminus \tilde{v}'} \|\nabla_{\mathbf{w}} \mathcal{L}_u\|_2 \\
&\leq 2|RF(\tilde{v})|C \\
&\leq 2\frac{K^{L+1} - 1}{K - 1}C
\end{aligned}
\tag{1}
$$

# B Algorithms

## B.1 DRW Sampler

The following algorithm shows how to generate disjoint subgraphs using the Disjoint Random Walks (DRW) sampling method (see Section 4.2). To generate a subgraph, we first sample a node $v$ from the set of remaining nodes, then remove it from this set. We then construct the set of valid neighbors of $v$, which consists of all nodes that have not been already sampled. We sample the next node $v$ in the subgraph from the set of valid neighbors, and repeat the process until we get a random walk of length $L$. We iterate this process until all nodes are included in one subgraph.

---

**Algorithm 2** DRW Sampler

---

  **Input:** Graph $G = \{V, E\}$, random walk length $L$.
  **Output:** Set of all disjoint subgraphs = ()
  remaining_nodes = $\{v_1, v_2, \ldots, v_N\}$
  **while** len(remaining_nodes) != 0 **do**
    subgraph = []
    v = sample(remaining_nodes, 1)            ▷ uniformly sample over non-sampled nodes
    subgraph.append(v)
    remaining_nodes.remove(v)
    l = 0
    **while** $l < L$ **do**
      valid_neighbors = Neighbors(v)         ▷ *Neighbors* returns all neighbors of a node
      **for** u in valid_neighbors **do**
        **if** u not in remaining_nodes **then**
          valid_neighbors.remove(u)
        **end if**
      **end for**
      **if** len(valid_neighbors) != 0 **then**
        v = sample(valid_neighbors, 1)        ▷ uniformly sample a neighbor of v
      **else**
        break
      **end if**
      random_walk.append(v)
      remaining_nodes.remove(v)
      $l = l + 1$
    **end while**
    subgraphs.add(subgraph)
  **end while**

---

## B.2 DRW-R Sampler

The following algorithm shows how to generate disjoint subgraphs using the Disjoint Random Walks with restarts (DRW-R) sampling method (see 4.2). The main difference to the DRW sampler is that, instead of stopping the subgraph generation after one random walk, we sample multiple random walks rooted at the same node by re-initializing the starting node of the random walk to the same root node of the subgraph $R$ times.

**Algorithm 3** DRW-R Sampler

---

**Input:** Graph $G = \{V, E\}$, random walk length $L$.
**Output:** Set of all disjoint subgraphs = ()
remaining_nodes = $\{v_1, v_2, \ldots, v_N\}$
**while** len(remaining_nodes) != 0 **do**
    subgraph = []
    root = sample(remaining_nodes, 1)                ▷ uniformly sample over non-sampled nodes
    subgraph.append(root)
    remaining_nodes.remove(root)
    **for** r in range(R) **do**
        v = root
        l = 0
        **while** $l < L$ **do**
            valid_neighbors = Neighbors(v)       ▷ *Neighbors* returns all neighbors of a node
            **for** u in valid_neighbors **do**
                **if** u not in remaining_nodes **then**
                    valid_neighbors.remove(u)
                **end if**
            **end for**
            **if** len(valid_neighbors) != 0 **then**
                v = sample(valid_neighbors, 1)         ▷ uniformly sample a neighbor of v
            **else**
                break
            **end if**
            subgraph.append(v)
            remaining_nodes.remove(v)
            $l = l + 1$
        **end while**
        subgraphs.add(subgraph)
    **end for**
**end while**

---

## C   Training Hyperparameters

We run all experiments with three different seeds, Adam optimizer and ReLU activation. We summarize the number of roots used for different sampling scenarios in Table 4. For the non-DP trainings, we fix the learning rate to 0.01. We perform a grid hyper-parameter search for the trainings on all datasets. We experiment with the following hyper-parameters for both DP and non-DP trainings:

- Number of layers in $\{1, 2\}$
- Width of hidden layers in $\{256, 512\}$
- Maximum graph degree in $\{2, 4\}$ for the FDP baseline

We use the follow hyper-parameters for the DP specific trainings:

- Learning rate in $\{0.01, 0.1, 0.2\}$
- Clip norm percentage $C\%$ in $\{0.001, 0.01, 0.1\}$.
- Noise multiplier $\lambda$ in $\{1, 2, 4, 8\}$. The noise multiplier is the ratio of the standard deviation $\sigma$ of the Gaussian noise added to the gradients to the sensitivity $\Delta_2 f$ of the function $f$. Instead of tuning $\sigma$, we tune $\lambda$, then fix $\sigma = \lambda \times \Delta_2 f$.
- Delta value $\delta$: we summarize the values used in Table 3

|          | Nodes   | Feature Size | Classes  | Training Nodes | Type         |
|----------|---------|--------------|----------|----------------|--------------|
| Cora     | 2,708   | 1,433        | 7 (s)    | 140            | Transductive |
| Citeseer | 3,327   | 3,703        | 6 (s)    | 120            | Transductive |
| PPI      | 14,755  | 50           | 121 (m)  | 9,716          | Inductive    |
| Pubmed   | 19,717  | 500          | 3 (s)    | 60             | Transductive |
| Flickr   | 89,250  | 500          | 7 (s)    | 44,625         | Inductive    |
| Arxiv    | 169,343 | 128          | 40 (s)   | 90,941         | Inductive    |
| Reddit   | 232,965 | 602          | 41 (s)   | 153,932        | Inductive    |

Table 2: Characteristics of the datasets that we use in our experiments. (s) indicates a single-label classification problem, and (m) a multi-label one.

|          | Dataset | | | | | | |
|----------|------|----------|------|--------|--------|-------|--------|
|          | Cora | Citeseer | PPI  | Pubmed | Flickr | Arxiv | Reddit |
| $\delta$ | 1e-5 | 1e-5     | 1e-5 | 1e-6   | 1e-6   | 1e-7  | 1e-7   |

Table 3: $\delta$ value used for each dataset.

| Sampler | Depth | Dataset | | | | | | |
|---------|-------|------|----------|-------|--------|--------|--------|--------|
|         |       | Cora | CiteSeer | PPI   | PubMed | Flickr | Arxiv  | Reddit |
| RW, uniform, PreDRW, | 1 | 70 | 60 | 2,000 | 30 | 10,000 | 20,000 | 30,000 |
| PreDRW-D, DynDRW     | 2 | 46 | 40 | 2,000 | 20 | 10,000 | 20,000 | 30,000 |
| PreDRW-R | 1 | 46 | 40 | 2,000 | 20 | 10,000 | 20,000 | 30,000 |
|          | 2 | 28 | 24 | 1,800 | 12 | 8,000  | 18,000 | 30,000 |

Table 4: Batch sizes used for training based on the sampler, depth of the model, and dataset. Note that as a general rule, we used around 20% of total number of training nodes for the large datasets, and 50% for the small datasets.

