# OpenReview forum: "Training Differentially Private Graph Neural Networks with Random Walk Sampling"
_NeurIPS.cc/2022/Workshop/TSRML — TSRML2022_

### Official Review · Reviewer_99ra · 2022-10-17

**Overall Rating:** 5

**Summary:**

The authors study the problem of training with differential privacy GNN. The usual approach of using DPSGD has a limitation that it is hard to give meaningful bounds for sensitivity in each batch. GNNs in fact have dependencies that reach over h hops and the number of nodes affected by a change in a user feature grows with deg^h in the worst case. The authors address this problem by using a random walk sampling approach. In their paper they create disjoint subgraphs and for each subgraph a node gradient is computed based only on the data within the subgraph. Since the subgraphs are disjoint a single change affects one subgraph. The rest of the privacy argument follow from amplification and composition. The authors show that this approach improves experimentally over prior work. From a model point of view, the authors work on feature privacy only and provide results for arbitrary GNN architectures


**Strengths:**

The paper offers an interesting take on the problem
The method seems to be scalable and practical

**Weaknesses:**

The paper is hard to follow. Key details of the paper are not presented well for instance how the training over the subgraphs works.
Most of the algorithms are in the appendix and not discussed in details.

The method works for feature privacy only and not graph privacy


**Overall Recommendation:**

Overall the paper is interesting but I found the algorithm to not be very clear.  Unfortunately, for lack of space, most of the technical work is not discussed and left in the appendix without comments.

One thing that is not clear to is how you obtain good embeddings on all nodes given that, it seems, for each subgraph only one node is trained (the starting point of the walk) and each subgraph is disjoint. This seems to suggest that you do training on only M nodes (the centers for the M subgraphs). How do you update the entire embedding table? Does this rely on rerunning the sampling independently many times?
Please provide more details on the algorithm, discuss in depths how you use the samples.



Minor: L45 gamma is not used in the definition of gamma-RDP
L82 should f depend upon X as well?
Fig 1 is hard to read for color blind people or in B&W print please use something else beside the color.





**Review Confidence:**

3: The reviewer is fairly confident that the evaluation is correct

---

### Official Review · Reviewer_72T6 · 2022-10-20
**Different sampling strategies for applying DP-SGD to Graph Neural Networks**

**Overall Rating:** 7

**Summary:**

The paper proposes a new method to train graph neural networks under DP. The method is based on random sampling of subgraphs of a given training graph. As mentioned, DP can be considered for "edge-level DP, where the edges between nodes are private; feature-level, where the features of a node are private; and node-level DP, where both the features and edges of a node are private". This work focuses on feature-level privacy and shows considerable improvements over state-of-the art for this.

This is obtained by dividing the graphs into disjoint subgraphs (dijoints sets of nodes) from which the DP-SGD sampling happens. This naturally gives the feature level DP and  it is straightforward to carry out the DP accounting for this subsampling method. Then, three random-walk based subsampling methods are considered and experimental results are given that show the benefits of this approach.

**Strengths:**

- The contribution seems clear and not simply a straightforward application of DP-SGD: there are actually three different random-walk strategies considered to generate the batches of nodes (the disjoint subgraphs).

- The paper is generally well written.

**Weaknesses:**


- The paper would benefit from more technical details on e.g. computing the gradients.

- The presentation regarding DP could be improved a bit. See e.g.:

p. 2, line 45: $\epsilon \rightarrow \gamma$

p.2 lines 53-54: sensitivity definition not correct, you should take supremum over all neighbouring datasets

You mention " Typically, S is assumed to be a Poisson or uniform sampling over the dataset." Then you don't explicitly mention which one you have used.

Remark:
please state more clearly in Table 1 that the first 2 methods are actually non-DP. At first glance, it is a bit confusing, it looks as if they were DP baselines.

**Overall Recommendation:**

Though GNNs are out of my area of expertise, I think this is a nice paper in a sense that it tailors DP-SGD to a problem in a non-trivial way by considering these different random-walk strategies to generate the batches. I am leaning towards acceptance.

**Review Confidence:**

3: The reviewer is fairly confident that the evaluation is correct

---

### Decision · Program_Chairs · 2022-10-23

**Decision:**

Accept

**Comment:**

The paper presents an interesting connection between privacy and graph neural networks. I hope the authors can improve the writing of this paper when preparing the final version as suggested by reviewers.